# OneBit: Towards Extremely Low-bit
# Large Language Models

**Yuzhuang Xu**[1]    **Xu Han**[2]    **Zonghan Yang**[2]    **Shuo Wang**[2]
**Qingfu Zhu**[1]    **Zhiyuan Liu**[2]    **Weidong Liu**[2]    **Wanxiang Che**[1,✉]
[1]Research Center for Social Computing and Information Retrieval,
Harbin Institute of Technology, Harbin, China
[2]Department of Computer Science & Technology, Tsinghua University, Beijing, China
`xyz21thu@gmail.com, car@ir.hit.edu.cn`

## Abstract

Model quantification uses low bit-width values to represent the weight matrices of existing models to be quantized, which is a promising approach to reduce both storage and computational overheads of deploying highly anticipated LLMs. However, current quantization methods suffer severe performance degradation when the bit-width is extremely reduced, and thus focus on utilizing 4-bit or 8-bit values to quantize models. This paper boldly quantizes the weight matrices of LLMs to 1-bit, paving the way for the extremely low bit-width deployment of LLMs. For this target, we introduce a 1-bit model compressing framework named OneBit, including a novel 1-bit parameter representation method to better quantize LLMs as well as an effective parameter initialization method based on matrix decomposition to improve the convergence speed of the quantization framework. Sufficient experimental results indicate that OneBit achieves good performance (at least 81% of the non-quantized performance on LLaMA models) with robust training processes when only using 1-bit weight matrices. Code and checkpoints are available at `https://github.com/xuyuzhuang11/OneBit`

## 1 Introduction

Transformer [36] has emerged as the pivotal architecture in large language models (LLMs), fundamentally reshaping the approach to natural language processing in deep learning era [6, 34, 4]. Despite their popularity, deploying transformer-based LLMs presents significant challenges due to their computational intensity and considerable memory requirements as the parameters of LLMs become more and more. For instance, even moderately-sized LLMs like LLaMA-13B [34] require around 26GB of memory to load its all parameters in FP16 format. Such overheads make deploying LLMs difficult beyond mid-to-high-end GPUs like the A100, let alone on mobile devices. The high demand for resources not only drives up usage costs, but also restricts their wider application.

Numerous efforts [10, 14, 13] have been devoted to reducing the computational and memory overheads of LLMs, while still preserving most of their original model capabilities. Among these efforts, quantization has gained widespread attention, particularly Post-Training Quantization (PTQ), benefitted from its lower transferring costs. Seminal studies such as GPTQ [14], SpQR [12], and AWQ [20] successfully compress the weight matrices of LLMs to 4-bit values while maintaining the main abilities of LLMs. Efficient quantization represents significant advances in LLM optimization, by achieving a balance between time and space efficiency as well as model performance.

Unfortunately, the efficacy of PTQ rapidly diminishes when the quantization bit-width is extremely low, as shown in Figure 1. Existing PTQ methods managed to compress weight matrices down to at least 3-bit [9]. Recent researches hope to leverage Quantization-Aware Training (QAT) to

38th Conference on Neural Information Processing Systems (NeurIPS 2024).

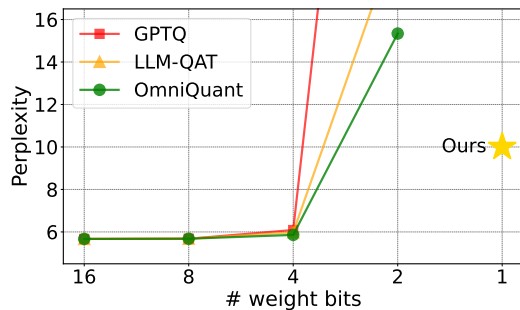

Figure 1: The perplexity (lower scores mean better performance) of existing widely-used low-bit quantization methods on LLaMA-7B, reported on Wikitext2 [23]. All the examined previous approaches suffer from significant performance degradation when quantizing models to 2-bit values. Our 1-bit quantization method can outperform these 2-bit baselines.

overcome the bottlenecks faced by PTQ. LLM-QAT [21] introduces a few learnable parameters into the quantization process, achieving notable results. OmniQuant [30], integrating learnable equivalent transformation, presents promising results in 2-bit quantization. However, existing methods decline when compressing model weights to 1 bit, struggling to maintain effectiveness. This mainly stems from the drastic precision loss at extremely low bit-width representation in weight matrix $\mathbf{W}$, significantly increasing loss in linear projection $\mathbf{WX}$, which is the core operator within LLMs.

In this paper, we propose a novel `Linear` layer and Sign-Value-Independent Decomposition (SVID) for weight matrices to represent LLMs using approximately 1-bit values. In our novel layer architecture, each original high-bit weight matrix is represented as one sign matrix ($\pm 1$) and two value vectors. The value vectors provide necessary floating-point precision in linear projection at little cost and help the model to be trained easily. The sign matrix maintains the high rank of the original weight matrix with a small space cost, thereby preserving high information capacity. SVID offers a better parameter initialization for 1-bit models from the non-quantized model and we employ quantization-aware knowledge distillation to transfer the capabilities of the original model to the proposed 1-bit counterpart. Experiments demonstrate that our method performs well at the W1A16 (1-bit weight and 16-bit activation) quantization level. Furthermore, our 1-bit model is more amenable to training and knowledge transfer than previous works. In summary, our contributions are 3-fold:

- We propose a novel and efficient 1-bit model architecture for LLMs, which can improve both the time and space efficiency during model inference. Moreover, our architecture is more stable during quantizing LLMs.

- We propose SVID to decompose high-bit matrices into low-bit ones, which is essential for the initialization of our 1-bit architecture. Experiments demonstrate that the SVID-based initialization can improve the model performance and convergence speed.

- Extensive experiments demonstrate that our method works well in model sizes from 1.3B to 13B in OPT, LLaMA, and LLaMA2, showcasing its generalizability.

## 2 Related Work

### 2.1 Large Language Model Compression

Quantization, pruning, and knowledge distillation (KD) are the mainstream methods for model compression. Quantization compresses model weights into low-bit values [14, 20, 11]. For data type alignment in computation and reducing memory, it also involves quantizing activation [10, 39] and key-value cache [30]. Pruning simplifies model complexity by removing unimportant weights or modules, thereby sparsifying the original larger models [13, 31, 22]. KD trains a smaller student model under the guidance of a larger teacher model [16, 1], achieving the purpose of compressing the larger one. Beyond these methods, low-rank factorization approximates the original weight matrix $\mathbf{W}$ with the product of two lower-rank matrices [40] and also achieves promising results. Our work belongs to quantization, using KD for knowledge transfer from the original LLM and uniquely

focusing on extremely low bit-width quantization. More details about model compression can refer to existing survies [37, 43].

## 2.2 Large Language Model Quantization

Since this paper aims to obtain extremely low-bit LLMs, here we thus introduce more details about LLM quantization. Quantization stands as a popular and crucial method for model compression, capable of achieving a significant compression ratio with a relatively small loss. It can be classified into Post-Training Quantization (PTQ) and Quantization-Aware Training (QAT) according to when quantization is applied.

PTQ directly converts trained models into lower-bit counterparts using accurate solvers and limited calibration data without additional training. Typically, GPTQ [14] row-wisely quantizes weight matrices and adjusts remaining weights to compensate for the precision loss caused by quantization, achieving nearly lossless 4-bit weight quantization. Moreover, numerous studies observed the effect of "outliers" in quantization [10, 18, 20]. LLM.int8() [10] suggests mixed-precision decomposition to ensure the accuracy of a few outliers in activations. SmoothQuant [39] reduces the difficulty of quantization by smoothing the outliers of activation. SpQR [12] identifies sensitive weights to ensure their precision, while quantizing other weights to lower bit-width.

QAT integrates quantization steps within the model, applying them during training or fine-tuning. It allows the model to better adapt to the reduced precision induced by quantization, leading to improved performance compared to PTQ. LLM-QAT [21] introduces a small number of learnable parameters into quantization and employs KD using data generated by the original model itself. OmniQuant (30; we classify it as QAT) further introduces learnable equivalent transformation, achieving acceptable results in 2-bit weight quantization. Contemporary work QuIP# [35] combines randomized Hadamard transform, vector quantization techniques, and fine-tuning to achieve better performance in 2-bit level. PEQA [17] and QLoRA [11] focus on fine-tuning a limited number of extra parameters to mitigate the precision loss caused by sub-4bit weight quantization. Our work is closely related to QAT, but due to the unique challenges posed by 1-bit quantization, our representation and initialization methods of quantized weights are distinct from any existing work.

## 3 Methodology

This section demonstrates our 1-bit architecture of the `Linear` layer to be quantized and discuss how to initialize the quantized model to achieve better performance in knowledge distillation. We start with a short review of classical weight quantization methods in Section 3.1 and then formulate our OneBit from Section 3.2 to Section 3.4 in detail.

### 3.1 Background

The main idea of model quantization is to compress each weight matrix $\mathbf{W}$ within models in FP32 or FP16 format to a low-bit counterpart. Specifically, we often quantize the weight matrices of `Linear` layers in transformer to 8, 4, and even 2 bits.

The majority of quantization studies primarily employ the round-to-nearest (RTN) method, by which the weight $w$ is rounded to the nearest value in the quantization grid. It can be formulated as

$$\hat{w} = \text{Clip}\left(\left\lfloor \frac{w}{s} \right\rceil + z, 0, 2^N - 1\right),\tag{1}$$

where $s$ denotes the quantization scale parameter, $z$ denotes the zero point parameter, and $N$ is the quantization bit-width. $\text{Clip}(\cdot)$ truncates the result in the range of 0 to $2^N - 1$. With the bit-width being lower and lower, the quantization grid also becomes sparser. When we quantize a LLM to 1-bit values, there are only 2 available numbers to be chosen in the quantized model. Existing study [9] points out that quantization based on the RTN method may get their best performance at the 4-bit level. Further quantizing to 2-bit values following this paradigm would result in a substantial degradation [30] as shown in Figure 1.

Furthermore, when $N$ equals 1, quantization based on RTN method is essentially equivalent to setting a threshold, with weight $w$ on either side of it being converted to corresponding integer value $\hat{w}$. In such a scenario, the parameters $s$ and $z$ in Eq. (1) effectively lose their practical

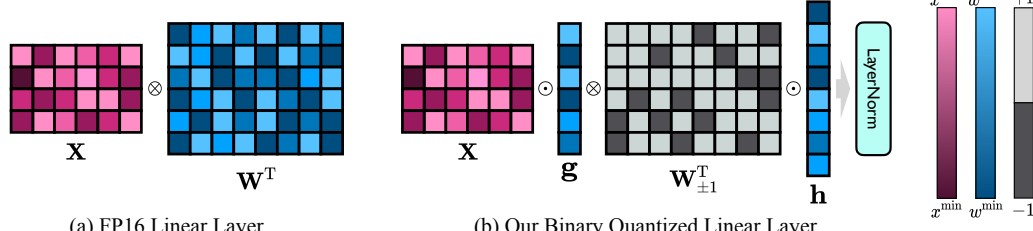


(a) FP16 Linear Layer      (b) Our Binary Quantized Linear Layer


Figure 2: The main idea of our method OneBit. The left is the original FP16 `Linear` Layer, in which both the activation $\mathbf{X}$ and the weight matrix $\mathbf{W}$ are in FP16 format. The right is our proposed architecture. Only value vectors $\mathbf{g}$ and $\mathbf{h}$ are in FP16 format, and the weight matrix consists of $\pm 1$ instead, which can be represented in INT1.

significance. Consequently, when quantizing weights to 1 bit, the element-wise RTN operation drastically undermines the precision of the weight matrix $\mathbf{W}$, leading to poor performance of the quantized model.

### 3.2 1-bit Linear Layer Architecture

Due to the severe precision loss of 1-bit weight quantization, converting weight matrices in `Linear` layers directly from FP32/16 to 1-bit format based on RTN is challenging. Wang et al. [38] explore this possibility by studying the capabilities of purely 1-bit weight matrices, training the 1-bit model *from scratch*. In the W1A16 setting, their `Linear` layers are designed as

$$\mathbf{W}_{\pm 1} = \text{Sign}\Big[\mathbf{W} - \text{Mean}(\mathbf{W})\Big],$$
$$\eta = \text{Mean}\Big[\text{Abs}\Big(\mathbf{W} - \text{Mean}(\mathbf{W})\Big)\Big], \tag{2}$$
$$\mathbf{Y} = \eta \cdot \text{LayerNorm}(\mathbf{X})\mathbf{W}_{\pm 1}^{\text{T}},$$

where $\mathbf{W}$ denotes the quantized weight matrix with the shape $m \times n$ and $\mathbf{W}_{\pm 1}$ denotes the 1-bit quantized matrix. $\mathbf{X}$ is the input of `Linear` layer and $\mathbf{Y}$ is the output. $\text{Sign}(\cdot)$, $\text{Mean}(\cdot)$ and $\text{Abs}(\cdot)$ functions return the sign matrix, average and absolute value matrix. Unfortunately, this approach reduces computational demands but also leads to a marked decrease in performance [38]. Moreover, due to training difficulties, experiments show that this method is challenging to use for quantizing existing models and can only be applied to training models *from scratch*.

Inspired by Wang et al. [38], we also quantize the weight matrix using the function $\text{Sign}(\cdot)$, and the element of the quantized matrix is set to +1 or -1 as well. Moreover, we also notice that although $\mathbf{W}_{\pm 1}$ maintains a high rank of $\mathbf{W}$, the missed floating-point precision still destroys the model performance. Therefore, different from previous work, we introduce 2 value vectors with an FP16 format to compromise the precision loss in the quantization process. During training, our proposed `Linear` layers are designed as

$$\mathbf{W}_{\pm 1} = \text{Sign}(\mathbf{W}),$$
$$\mathbf{Y} = \big[(\mathbf{X} \odot \mathbf{g})\mathbf{W}_{\pm 1}^{\text{T}}\big] \odot \mathbf{h}, \tag{3}$$
$$\mathbf{Z} = \text{LayerNorm}(\mathbf{Y}),$$

where $\mathbf{g}$ and $\mathbf{h}$ are the two FP16 value vectors. During inference, $\mathbf{W}_{\pm 1}$ is packed with an INT1 format, and $\text{Sign}(\cdot)$ will not be used, as shown in Figure 2. Note that we specify the calculation order using brackets in Eq. (3) for minimizing the time and space cost. The main difference between Wang et al. [38] and OneBit is the extra parameter $\mathbf{g}$ and $\mathbf{h}$. Even if additional parameters are brought in, the benefits far outweigh its small cost. For instance, when we quantize one weight matrix with the shape $4096 \times 4096$, the average bit-width of the quantized result is *1.0073*. See A.7 for the details.

### 3.3 Sign-Value-Independent Decomposition

In our proposed 1-bit architecture, the weight matrix $\mathbf{W}$ is mathematically divided into two components: one sign matrix $\mathbf{W}_{\pm 1}$ in INT1 format and two value vector $\mathbf{g/h}$ in FP16 format. To initialize

the 1-bit model with the help of the fully trained weight, we introduce the Sign-Value-Independent Decomposition (SVID) of the weight matrix $\mathbf{W}$, which can be formulated as $\mathbf{W} = \mathbf{W}_{\text{sign}} \odot \mathbf{W}_{\text{value}}$. Here we have $\mathbf{W}_{\text{value}} = |\mathbf{W}|$ and $\mathbf{W}_{\text{sign}} = \text{Sign}(\mathbf{W})$. For $\mathbf{W}_{\text{value}}$, we further approximately decompose it into the outer product of two vectors $\mathbf{a}$ and $\mathbf{b}$, which is also known as *rank-1 approximation*. Hence, our proposed matrix decomposition method can be represented as

$$\mathbf{W} \approx \mathbf{W}_{\text{sign}} \odot \left(\mathbf{a}\mathbf{b}^{\text{T}}\right). \tag{4}$$

We can employ some widely used matrix decomposition methods to perform the rank-1 approximation, such as SVD [2] and NMF [25].

**Proposition 1**  Given the weight matrix $\mathbf{W}$ and input $\mathbf{X}$, the `Linear` layer can be reformulated as the following according to SVID:

$$\mathbf{X}\mathbf{W}^{\text{T}} \approx \left[\left(\mathbf{X} \odot \mathbf{b}^{\text{T}}\right)\mathbf{W}_{\text{sign}}^{\text{T}}\right] \odot \mathbf{a}^{\text{T}}. \tag{5}$$

We prove this approximation in Appendix A.1. This bridges the gap between the architecture of the quantized model and its original weights. It indicates that if we assign $\mathbf{W}_{\text{sign}}$ to $\mathbf{W}_{\pm 1}$, $\mathbf{a}^{\text{T}}$ to $\mathbf{h}$ and $\mathbf{b}^{\text{T}}$ to $\mathbf{g}$, the quantized model is an approximate initialization of the original model. Moreover, compared to restoring the original matrix $\mathbf{W}$ first (such as in Eq. (4)), the computational order in Eq. (5) saves approximately one matrix $\mathbf{W}$ in FP16 format in memory as there is no need to restore $\mathbf{W}$ in FP16 format.

The main objective of SVID is to involve the sign matrix $\mathbf{W}_{\text{sign}}$ in approximating matrix $\mathbf{W}$, rather than solely relying on value vectors in FP16 format. To substantiate the role of the sign matrix $\mathbf{W}_{\text{sign}}$ in matrix approximation, we present the following proposition.

**Proposition 2**  Given matrices $\mathbf{W}$ and $|\mathbf{W}|$, $\mathbf{W} = \mathbf{W}_{\text{sign}} \odot |\mathbf{W}|$. We decompose these matrices in the way $\mathbf{W} = \mathbf{a}\mathbf{b}^{\text{T}} + \mathbf{E}_1$ and $|\mathbf{W}| = \tilde{\mathbf{a}}\tilde{\mathbf{b}}^{\text{T}} + \mathbf{E}_2$, where $\mathbf{E}_i$ denotes the error matrices. In terms of the Frobenius-norm, the SVID is closer to the original matrix $\mathbf{W}$:

$$\left\|\mathbf{W} - \mathbf{W}_{\text{sign}} \odot \tilde{\mathbf{a}}\tilde{\mathbf{b}}^{\text{T}}\right\|_{\text{F}}^2 \leq \left\|\mathbf{W} - \mathbf{a}\mathbf{b}^{\text{T}}\right\|_{\text{F}}^2. \tag{6}$$

We also prove this proposition in Appendix A.1. It clearly demonstrates the practical role of the sign matrix $\mathbf{W}_{\text{sign}}$ in matrix approximation.

Note that, given the predominantly low precision of most parameters, it is quite challenging to approximate the weight matrix $\mathbf{W}$ accurately. SVID is not aimed to precisely replicate the original model's parameters, but to provide an effective starting point for further training, leveraging the extensive training of the original model. Details on transferring knowledge from the original model to the quantized counterpart are in Section 3.4.

### 3.4  Knowledge Transfer

We employ quantization-aware knowledge distillation to transfer knowledge from the original model (i.e. teacher model) to the quantized one (i.e. student model). In the student model, the element in matrix $\mathbf{W}$ and vectors $\mathbf{g}/\mathbf{h}$ in Eq. (3) will be trained. We use cross-entropy based logits and mean-square-error based hidden state of the full-precision teacher model to direct the quantized student model [32]. Language modeling loss is not used. The cross-entropy is defined as

$$\mathcal{L}_{\text{CE}} = -\frac{1}{n_s}\sum_{i=1}^{n_s}\sum_c P_c^{\mathcal{T}}\left(\mathbf{o}_i\right)\log P_c^{\mathcal{S}}\left(\mathbf{o}_i\right), \tag{7}$$

where $c$ denotes the number of classes and $n_s$ denotes the number of training samples in the current batch. $\mathcal{T}$ and $\mathcal{S}$ are the teacher model and student model, respectively. The error of hidden states is defined as

$$\mathcal{L}_{\text{MSE}} = \sum_{i=1}^{n_s}\sum_{j=1}^{n_l}\left\|\frac{\mathbf{q}_{i,j}^{\mathcal{T}}}{\left\|\mathbf{q}_{i,j}^{\mathcal{T}}\right\|_2} - \frac{\mathbf{q}_{i,j}^{\mathcal{S}}}{\left\|\mathbf{q}_{i,j}^{\mathcal{S}}\right\|_2}\right\|_2^2, \tag{8}$$

where $n_l$ denotes the number of layers and $\mathbf{q}$ denotes the hidden state. Hence the final objective function can be formulated as

$$\mathcal{L}_{\text{KD}} = \mathcal{L}_{\text{CE}} + \alpha\mathcal{L}_{\text{MSE}}, \tag{9}$$

where $\alpha$ is the hyper-parameter that balances the importance of the cross-entropy loss and the features in the intermediate layers. Please refer to A.6 for further discussions of this part.

## 4 Experiments

We experiment with 1-bit weight-only quantizaton and maintain 16-bit activation (W1A16) in this work. We evaluate our approach by performing experiments on OPT-1.3B/2.7B models, LLaMA-7B/13B models and LLaMA2-7B/13B models, and present results on various tasks.

### 4.1 Settings

**Data** For the training data of our quantization-aware knowledge distillation, we follow Liu et al. [21] to synthesize corpus using next token generation from the original teacher model. It randomizes the first token from vocabulary and generates the next token iteratively until reaching either the *<EOS>* token or the maximum length. Specially, the top-1 predictions are selected deterministically for the first 3 to 5 tokens, followed by stochastic sampling for the remaining tokens. We utilized LLaMA-7B to generate a total of 132k data entries, each with a maximum length of 2,048.

**Training Details** Every KD experiment learns the training data over 50 epochs, from which 2048-token segments are selected. We employ NMF in scikit-learn [1] to decompose the weight matrices in SVID. The quantized student models are optimized by Adam [19] with $\beta_1 = 0.9$, $\beta_2 = 0.98$. The learning rate for all experiments is scheduled by *cosine* strategy. We use NVIDIA A100 GPUs and maintain FP16 precision while training quantized models. For additional details such as learning rate, please refer to Table 1.

Table 1: Training details of knowledge distillation.

| Models | learning rate | $\alpha$ | # GPUs |
|---|---|---|---|
| OPT-1.3B | 4e-4 | 1.0 | $1 \times 8$ |
| OPT-2.7B | 2e-4 | 1.0 | $1 \times 8$ |
| LLaMA-7B | 4e-4 | 1.0 | $1 \times 8$ |
| LLaMA-13B | 2e-4 | 1.0 | $2 \times 8$ |
| LLaMA2-7B | 1e-4 | 1.0 | $1 \times 8$ |
| LLaMA2-13B | 2e-4 | 1.0 | $2 \times 8$ |

**Baselines** To our knowledge, there is no previous work exploring the 1-bit quantization of LLMs from a knowledge transfer perspective. To this end, we relax the quantization bit-width of baselines to 2 bits (*W2A16*) while maintaining the *W1A16* setting in our method. We compare our method with GPTQ [14], LLM-QAT [21] and OmniQuant [30]. To ensure a fair comparison in terms of space usage, baselines do not employ grouped quantization. Additionally, we included the results of vanilla transformers with FP16 precision as a reference. While the recent work BitNet [38] also introduced one 1-bit model architecture, it only worked for training models from scratch. We also analyze its capability to transfer knowledge from the original models in Appendix A.5.

**Evaluation Metrics** Basically, we evaluate quantized models by testing the perplexity on the validation set, specifically on WikiText2 [23] and C4 [28]. Lower perplexity indicates that the compressed model is better at preserving the output distribution of the original model. Furthermore, accuracies of zero-shot tasks including Winogrande [29], HellaSwag [41], PIQA [4], BoolQ [7], and ARC [8] are also reported. They evaluate if the capabilities of the original model on downstream tasks are retained. We utilize the open-sourced toolkit "LM-Evaluation-Harness"[2] to perform the perplexity test and all zero-shot tasks.

---

[1] https://scikit-learn.org/
[2] https://github.com/EleutherAI/lm-evaluation-harness

Table 2: Main results of evaluation experiment. We report the perplexity and zero-shot accuracy. "FP16" is the transformer with FP16 parameters and we refer to it as the upper-bound of all the methods. The **best** score is bolded.

| Models | Methods | Perplexity(↓) | | Zero-shot Accuracy(↑) | | | | | | |
| | | Wiki2 | C4 | Wino. | Hella. | PIQA | BoolQ | ARC-e | ARC-c | Avg. |
|---|---|---|---|---|---|---|---|---|---|---|
| OPT-1.3B | FP16 | 14.63 | 14.72 | 59.67 | 53.73 | 72.42 | 57.68 | 50.80 | 29.69 | 54.00 |
| | GPTQ | 9.5e3 | 3.8e3 | 49.33 | 25.57 | 52.07 | 39.60 | 26.68 | 23.63 | 36.15 |
| | LLM-QAT | 4.9e3 | 2.1e3 | 49.72 | 25.72 | 50.05 | 37.83 | 25.76 | **25.09** | 35.70 |
| | OmniQuant | 42.43 | 55.64 | **51.85** | 33.39 | 60.94 | 56.45 | 38.76 | 23.38 | 44.13 |
| | OneBit | **25.42** | **22.95** | 51.14 | **34.26** | **62.57** | **59.45** | **41.25** | 24.06 | **45.46** |
| OPT-2.7B | FP16 | 12.47 | 13.17 | 60.93 | 60.59 | 74.81 | 60.28 | 54.34 | 31.31 | 57.04 |
| | GPTQ | 8.7e3 | 3.9e3 | 49.88 | 26.47 | 49.84 | 39.88 | 25.76 | **26.02** | 36.31 |
| | LLM-QAT | 3.7e3 | 1.4e3 | 52.09 | 25.47 | 49.29 | 37.83 | 24.92 | 25.60 | 35.87 |
| | OmniQuant | 30.25 | 41.31 | 51.62 | **38.21** | 62.19 | 54.25 | 40.82 | 24.74 | 45.31 |
| | OneBit | **21.86** | **20.76** | 51.67 | 38.18 | **63.87** | 54.28 | **43.39** | 24.40 | **45.97** |
| LLaMA-7B | FP16 | 5.68 | 7.08 | 66.85 | 72.99 | 77.37 | 73.21 | 52.53 | 41.38 | 64.06 |
| | GPTQ | 1.9e3 | 7.8e2 | 49.41 | 25.63 | 49.95 | 43.79 | 25.84 | 27.47 | 37.02 |
| | LLM-QAT | 7.1e2 | 3.0e2 | 51.78 | 24.76 | 50.87 | 37.83 | 26.26 | 25.51 | 36.17 |
| | OmniQuant | 15.34 | 26.21 | 52.96 | 43.68 | 62.79 | **58.69** | 41.54 | 29.35 | 48.17 |
| | OneBit | **10.19** | **11.40** | **58.48** | **51.54** | **68.01** | 57.28 | **42.47** | **30.20** | **51.33** |
| LLaMA-13B | FP16 | 5.09 | 6.61 | 70.17 | 76.24 | 79.05 | 68.47 | 59.85 | 44.54 | 66.39 |
| | GPTQ | 3.2e3 | 9.9e2 | 50.67 | 25.27 | 50.00 | 42.39 | 26.14 | 27.39 | 36.98 |
| | LLM-QAT | 1.8e3 | 1.2e3 | 51.62 | 25.40 | 50.33 | 37.83 | 27.02 | 26.87 | 36.51 |
| | OmniQuant | 13.43 | 19.33 | 53.83 | 54.16 | 68.99 | 62.20 | **45.50** | 30.38 | 52.51 |
| | OneBit | **9.18** | **10.25** | **62.90** | **56.78** | **70.67** | **64.16** | 44.53 | **32.00** | **55.17** |

## 4.2 Main Results

Table 2 compares our method with other typical strong baselines on different models. Due to space limitations, results of LLaMA2-7B/13B are listed in Appendix A.3. In various model sizes, our 1-bit weight quantization method obviously outperforms others under the W2A16 setting. Moreover, the effectiveness of QAT based methods consistently improves as the model size increases, whereas the result of the PTQ method, GPTQ, may degrade when model size increases (e.g., from 7B to 13B on LLaMA). This demonstrates that QAT-based method can achieve stable results in extremely low-bit quantization. Specifically, our method approaches the performance of FP16 more closely as the model size increases. For instance, when scaling from LLaMA-7B to LLaMA-13B, the perplexity (on C4) of the FP16 model decreases by only 0.47, whereas our method sees a reduction of 1.15.

For perplexity, only our method achieves comparable results to the strongest FP16 baseline. For instance, our method achieves 9.18 in the Wiki2 dataset on LLaMA-13B model and the FP16 baseline is 5.09. The performance loss of other methods is significant, even though they use 2-bit quantization, which is more than our 1 bit. For GPTQ and LLM-QAT, the performance degradation after quantization is pretty severe. As for OmniQuant, even though it is the strongest baseline under the W2A16 setting, it still suffers greater performance loss compared to our W1A16 setting.

For zero-shot accuracy, although all methods inevitably have some degradation, our method achieves the closest performance to the FP16 baseline among most models. On the OPT-1.3B/2.7B model, our method shows smaller performance loss on most tasks such as PIQA and ARC-e. Additionally, the loss of other tasks is negligible compared with the second-best baseline, OmniQuant. On the LLaMA-7B model, our method also notably outperforms OmniQuant in most tasks except BoolQ/ARC-e, averaging about a 4% improvement overall.

## 4.3 Problem Solving Ability

We have demonstrated the superior performance of our method under the W1A16 setting, compared to other representative baselines. Although all methods inevitably face performance degradation in

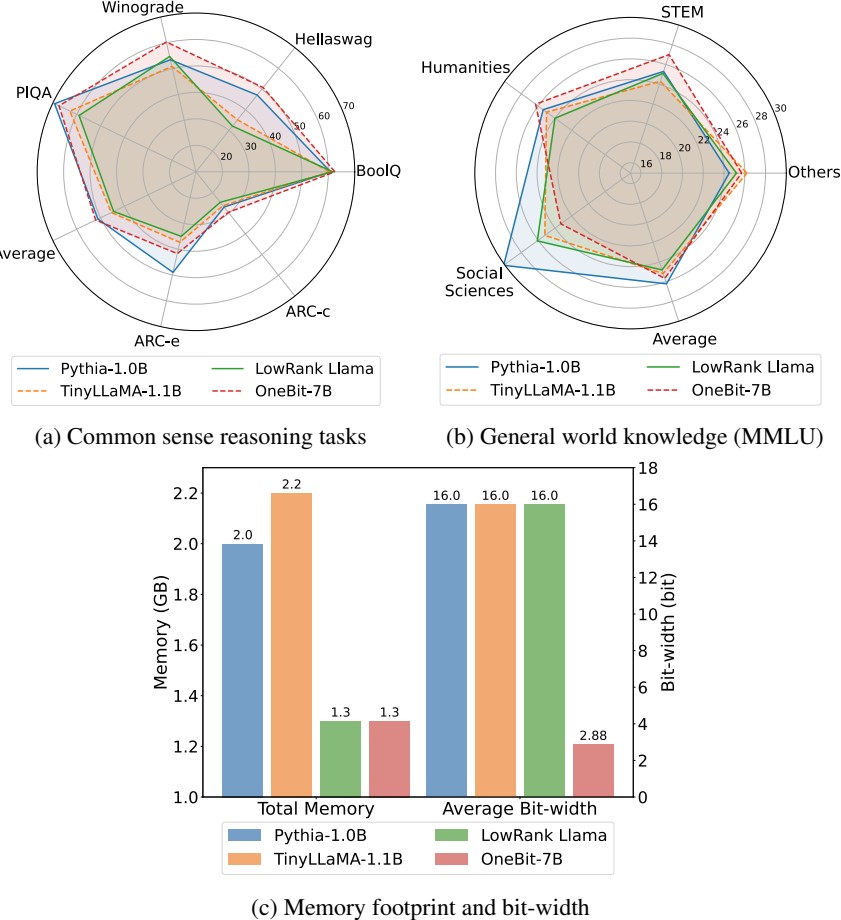

(a) Common sense reasoning tasks

(b) General world knowledge (MMLU)

(c) Memory footprint and bit-width

Figure 3: Comparison of model capabilities and compressive degree.

1-bit weight quantization, it remains of interest how our method fares in solving practical problems among the various approaches to reducing model size. For instance, directly training smaller models [42] or employing low-rank decomposition to reduce the number of parameters.

To this end, we consider two crucial abilities of LLMs: commonsense reasoning and world knowledge. For commonsense reasoning, we use the 6 tasks (Hellaswag, etc.) and settings described in Section 4.2. For world knowledge, we examine it using the Massive Multi-task Language Understanding (MMLU; 15), a benchmark that covers wide domains and knowledge. We compare the following 4 models:

**Pythia-1.0B** [3] A well-trained model released by EleutherAI whose memory footprint is 1.54x that of our OneBit-7B model.

**TinyLLaMA-1.1B** [42] A model with the same structure as the LLaMA models, which undergoes continued training. To compare fairly, we use the checkpoint at 10k training steps, which is 2x that of our OneBit-7B model.

**LowRank LLaMA** [24] Decompose every weight matrix in `Linear` layers to two low-rank matrices and learn from the original LLaMA-7B model by KD in the same setting of OneBit-7B.

**OneBit-7B** The model that we use in Section 4.2, which is built with OneBit.

Figure 3a and 3b demonstrate common sense reasoning ability and general world knowledge of different models. We can observe that, although other models have more parameters and are more thoroughly trained than ours, our model still has advantages in common sense reasoning. This reflects the benefits inherited from the larger 7B model. In terms of world knowledge, despite a significant loss in social sciences, our model outperforms the fully trained Pythia-1B in other domains. These results demonstrate the practical usability of OneBit.

Table 3: Compression ratio of LLaMA models.

| Models | FP16 (GB) | OneBit (GB) | Ratio (%) |
|--------|-----------|-------------|-----------|
| LLaMA-7B | 13.5 | 1.3 | 90.4 |
| LLaMA-13B | 26.0 | 2.2 | 91.5 |
| LLaMA-30B | 65.1 | 4.9 | 92.5 |
| LLaMA-65B | 130.6 | 9.2 | 93.4 |

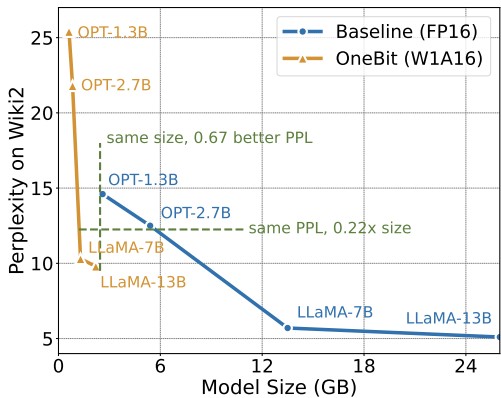

Figure 4: Tradeoff between size and PPL.

# 5 Analysis and Discussion

## 5.1 Efficiency

It is evident that extremely low-bit quantization of weights can significantly reduce the memory footprint of models. As shown in Table 3, the actual compression ratio increases as the model size increases. This is particularly meaningful for larger models, making it possible to fit the model into one GPU. While there is a performance loss, Figure 4 illustrates that our method achieves a good trade-off between space occupancy and model performance. For example, we can achieve comparable performance to FP16 with only 0.2x the model space. Furthermore, quantizing to $\pm 1$ also aids in accelerating matrix multiplication on CPUs. It is because the floating-point multiplication of elements in two matrices can be converted into much faster bit operations on these chips. Thus the substantial reduction in memory overhead makes these low-bit LLMs meet the requirements for deployment on PCs and smartphones.

## 5.2 Robustness

Existing work [38] has already noted the instability within QAT. Extremely low-bit quantization makes the training process highly sensitive to the learning rate, making it difficult for the model to converge when the rate is too small or too large. This is primarily due to the large magnitude of gradients generated as the weight elements fluctuate between +1 and -1, leading to substantial fluctuations in the output of `Linear` layers. Experiments demonstrate that OneBit shows more stable training process and is not sensitive to learning rates. Please refer to Appendix A.5 for more details.

## 5.3 Effect of Different Components

The variable components in our method primarily include Post-LayerNorm, value vectors, and parameter initialization.

**Post-LayerNorm** We discover that there might be floating-point overflow during the QAT process. As depth increases, the activation can become progressively larger. We tackle it using Post-LayerNorm instead of Pre-LayerNorm. In contrast, Pre-LayerNorm may occasionally be ineffective.

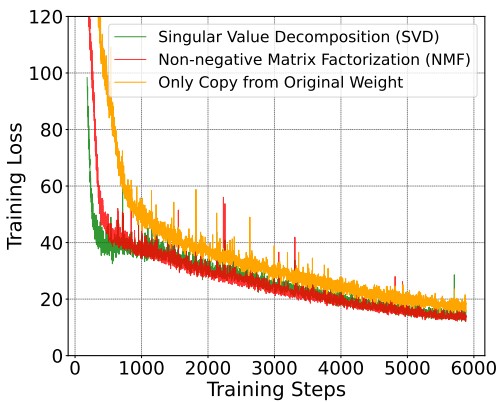

Figure 5: Training process of OneBit-7B.

**Value Vectors** The main structural difference between OneBit and BitNet [38] is the two value vectors, which are demonstrated to be effective in Section 4.2. They facilitate stable training and the knowledge transfer process. Please refer to Appendix A.5 for more details of comparison.

**Parameter Initialization** In our proposed SVID, both NMF and SVD can be used to decompose $|\mathbf{W}|$ and we recommend using the former. This is because we find that NMF may make the training more faster to converge. Figure 5 shows that initializing by NMF facilitates better performance.

## 6 Conclusion

We propose a novel model structure for 1-bit weight quantization and a corresponding parameter initialization method to address the difficulty in 1-bit quantization. Extensive experiments on LLMs of various sizes and series demonstrate that OneBit has clear advantages over representative strong baselines and achieves a good tradeoff between model size and performance. We further analyze the capabilities of such extremely low-bit quantized models and provide guidance for future research.

## Acknowledgments

We gratefully acknowledge the support of the National Natural Science Foundation of China (NSFC) via grant 62236004, 62206078, 62441603 and 62476073. This work was also supported by the National Key Research and Development Program of China under grant 2023YFB4503000.

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

# A  Appendix

## A.1  Proofs of Propositions

In this section, we provide the necessary and detailed proofs for the propositions presented in this paper. All symbols have the same definition as in the main text.

**Proposition 1**  Given the weight matrix $\mathbf{W}$ and input $\mathbf{X}$, the `Linear` layer can be reformulated as the following according to SVID:

$$\mathbf{X}\mathbf{W}^{\mathrm{T}} \approx \left[ \left( \mathbf{X} \odot \mathbf{b}^{\mathrm{T}} \right) \mathbf{W}_{\mathrm{sign}}^{\mathrm{T}} \right] \odot \mathbf{a}^{\mathrm{T}}.$$

**Proof**  From Eq. (4), we have $w_{ij} \approx s_{ij} \cdot a_i b_j$, where $s_{ij}$ is the element of $\mathbf{W}_{\mathrm{sign}}$. Hence we have

$$
\begin{aligned}
\left( \mathbf{X}\mathbf{W}^{\mathrm{T}} \right)_{ij} &\approx \sum_k x_{ik} w_{kj}^{\mathrm{T}} = \sum_k x_{ik} w_{jk} \\
&= \sum_k x_{ik} s_{jk} a_j b_k \\
&= \sum_k x_{ik} b_k s_{jk} a_j \\
&= \sum_k \left( \mathbf{X} \odot \mathbf{b}^{\mathrm{T}} \right)_{ik} s_{kj}^{\mathrm{T}} a_j \\
&= \left[ \left( \mathbf{X} \odot \mathbf{b}^{\mathrm{T}} \right) \mathbf{W}_{\mathrm{sign}}^{\mathrm{T}} \right]_{ij} a_j \\
&= \left\{ \left[ \left( \mathbf{X} \odot \mathbf{b}^{\mathrm{T}} \right) \mathbf{W}_{\mathrm{sign}}^{\mathrm{T}} \right] \odot \mathbf{a}^{\mathrm{T}} \right\}_{ij}.
\end{aligned}
$$

This proposition is proved.

**Lemma 1**  Let $\sigma_i \left( \mathbf{W} \right)$ denote the $i$-th biggest singular value of matrix $\mathbf{W}$. The following inequality holds:

$$\sigma_1 \left( |\mathbf{W}| \right) \geq \sigma_1 \left( \mathbf{W} \right).$$

**Proof**  According to the definition of induced norm, there are

$$\sigma_1 \left( \mathbf{W} \right) = \|\mathbf{W}\|_2 = \max_{\mathbf{x}, \|\mathbf{x}\|_2 = 1} \|\mathbf{W}\mathbf{x}\|_2,$$

$$\sigma_1 \left( |\mathbf{W}| \right) = \||\mathbf{W}|\|_2 = \max_{\mathbf{y}, \|\mathbf{y}\|_2 = 1} \||\mathbf{W}|\mathbf{y}\|_2.$$

Note that for $\forall \mathbf{x}, \|\mathbf{x}\|_2 = 1$ and we have

$$
\begin{aligned}
\||\mathbf{W}||\mathbf{x}|\|_2^2 &= \sum_i \left( \sum_j |w_{ij}||x_j| \right)^2 \\
&\geq \sum_i \left( |\sum_j w_{ij} x_j| \right)^2 \\
&= \sum_i \left( \sum_j w_{ij} x_j \right)^2 = \|\mathbf{W}\mathbf{x}\|_2^2.
\end{aligned}
$$

Therefore

$$\max_{\mathbf{y}, \|\mathbf{y}\|_2 = 1} \||\mathbf{W}|\mathbf{y}\|_2 \geq \max_{\mathbf{x}, \|\mathbf{x}\|_2 = 1} \|\mathbf{W}\mathbf{x}\|_2.$$

This lemma is proved.

**Proposition 2** Given matrices $\mathbf{W}$ and $|\mathbf{W}|$, $\mathbf{W} = \mathbf{W}_{\text{sign}} \odot |\mathbf{W}|$. We decompose these matrices in the way $\mathbf{W} = \mathbf{a}\mathbf{b}^{\text{T}} + \mathbf{E}_1$ and $|\mathbf{W}| = \tilde{\mathbf{a}}\tilde{\mathbf{b}}^{\text{T}} + \mathbf{E}_2$, where $\mathbf{E}_i$ denotes the error matrices. In terms of the Frobenius-norm, the SVID is closer to the original matrix $\mathbf{W}$:

$$\left\| \mathbf{W} - \mathbf{W}_{\text{sign}} \odot \tilde{\mathbf{a}}\tilde{\mathbf{b}}^{\text{T}} \right\|_{\text{F}}^2 \leq \left\| \mathbf{W} - \mathbf{a}\mathbf{b}^{\text{T}} \right\|_{\text{F}}^2 .$$

**Proof** Here we consider SVD to prove it. For SVD, the norm of the error matrix $\mathbf{E}$ in the rank-1 approximation is the sum of the squares of all singular values except for the largest one. We have

$$\|\mathbf{E}_1\|_F^2 = \sum_{i=2}^{n} \sigma_i^2\left(\mathbf{W}\right),$$

$$\|\mathbf{E}_2\|_F^2 = \sum_{i=2}^{n} \sigma_i^2\left(|\mathbf{W}|\right).$$

Based on $\|\mathbf{W}\|_F^2 = \||\mathbf{W}|\|_F^2$, we have

$$\sum_{i=1}^{n} \sigma_i^2\left(\mathbf{W}\right) = \sum_{i=1}^{n} \sigma_i^2\left(|\mathbf{W}|\right).$$

According to Lemma 1, we can conclude

$$\|\mathbf{E}_2\|_F^2 \leq \|\mathbf{E}_1\|_F^2.$$

From the equation in this proposition, we can formulate

$$\mathbf{W}_{\text{sign}} \odot |\mathbf{W}| = \mathbf{W}_{\text{sign}} \odot \tilde{\mathbf{a}}\tilde{\mathbf{b}}^{\text{T}} + \mathbf{W}_{\text{sign}} \odot \mathbf{E}_2.$$

Hence we have

$$\mathbf{W} - \mathbf{W}_{\text{sign}} \odot \tilde{\mathbf{a}}\tilde{\mathbf{b}}^{\text{T}} = \mathbf{W}_{\text{sign}} \odot \mathbf{E}_2.$$

Therefore

$$\|\mathbf{W}_{\text{sign}} \odot \mathbf{E}_2\|_F^2 = \sum_{i,j} s_{ij}^2 e_{ij}^2 = \sum_{i,j} e_{ij}^2$$
$$= \|\mathbf{E}_2\|_F^2 \leq \|\mathbf{E}_1\|_F^2,$$

where $s_{ij} = \pm 1$ is the element of $\mathbf{W}_{\text{sign}}$. Hence the inequation in this proposition is proved.

### A.2 Details on Baselines

In this subsection, we provide the essential details of the baselines in this work:

- GPTQ [14]: We employ the open-source code released by the author. Both OPT models and LLaMA models take 128 2048-token samples from the C4 dataset to calibrate the quantized model. For LLaMA models, we apply the activation order heuristic according to the recommendation from the code.

- LLM-QAT [21]: We reimplement this method to adapt the W2A16 setting, as LLM-QAT is not designed for 2-bit weight quantization. We also do not quantize the KV Cache. When quantizing the weight matrix in `Linear` layer, we use symmetric MinMax quantization in which the zero-point is set to 0. The training hyper-parameters are the same as ours. Please refer to the training details in Section 4.1.

- OmniQuant [30]: We employ the open-source code released by the author. Both OPT models and LLaMA models take 128 2048-token samples from the WikiText2 dataset to calibrate the quantized model. The learning rate for learnable weight clipping and equivalent transformation is set to 5e-3 and 1e-2, respectively. We use a batch size of 1 and train 40 epochs for each model. For OPT models, both learnable weight clipping and equivalent transformation are leveraged. For LLaMA models, only learnable weight clipping is used.

## A.3 Results of LLaMA2

Table 4 compares the results on LLaMA2-7B/13B. Obviously, our method has advantages in both perplexity and zero-shot accuracy. It also reflects that the advantages of our method are more pronounced in larger models. For instance, when scaling from LLaMA2-7B to LLaMA2-13B, the perplexity of the FP16 model decreases by around only 0.5, whereas our method reduces it by around 1.0 on both Wiki2 and C4 datasets.

Table 4: Results of LLaMA2. We bold the **best** scores.

| Models | Methods | Perplexity($\downarrow$) | | Zero-shot Accuracy($\uparrow$) | | | | | | |
| | | Wiki2 | C4 | Wino. | Hella. | PIQA | BoolQ | ARC-e | ARC-c | Avg. |
|---|---|---|---|---|---|---|---|---|---|---|
| LLaMA2-7B | FP16 | 5.47 | 6.97 | 67.09 | 72.94 | 76.88 | 71.10 | 53.58 | 40.61 | 63.70 |
| | GPTQ | 7.7e3 | NAN | 50.28 | 26.19 | 49.46 | 42.97 | 26.77 | 28.58 | 37.38 |
| | LLM-QAT | 1.1e3 | 6.6e2 | 49.08 | 25.10 | 50.12 | 37.83 | 26.26 | 26.96 | 35.89 |
| | OmniQuant | 31.21 | 64.34 | 51.22 | 33.87 | 56.53 | 59.14 | 33.63 | 24.32 | 43.12 |
| | OneBit | **9.73** | **11.11** | **58.41** | **52.58** | **68.12** | **63.06** | **41.58** | **29.61** | **52.23** |
| LLaMA2-13B | FP16 | 4.88 | 6.47 | 69.77 | 76.62 | 79.05 | 68.99 | 57.95 | 44.20 | 66.10 |
| | GPTQ | 2.1e3 | 3.2e2 | 51.85 | 25.67 | 51.74 | 40.61 | 25.46 | 27.30 | 37.11 |
| | LLM-QAT | 5.1e2 | 1.1e3 | 51.38 | 24.37 | 49.08 | 39.85 | 27.15 | 24.32 | 36.03 |
| | OmniQuant | 16.88 | 27.02 | 53.20 | 50.34 | 62.24 | 62.05 | 40.66 | 29.61 | 49.68 |
| | OneBit | **8.76** | **10.15** | **61.72** | **56.43** | **70.13** | **65.20** | **43.10** | **33.62** | **55.03** |

## A.4 Instruction Following Ability

Instruction following is an important ability of LLMs [27, 5, 26]. Beyond the discussion on model abilities and efficiency before, we also focus on the instruction following ability of extremely low-bit models, which is closely related to their practical usability. In this subsection, we empirically study this capability of our quantized model. We fine-tune the model for 3 epochs using the alpaca_en_52k dataset and alpaca templates [33], then observe the generation in both zero-shot and few-shot settings before and after fine-tuning. During training, the learning rate is set to 1e-7 and the batch size to 32. Other parameters are consistent with Section 4.1.

Table 5 demonstrates the content generation and instruction following abilities of our 7B model. Under the zero-shot setting, the model without SFT produced verbose, repetitive, and low-quality text. However, once experienced to SFT, our model is able to smoothly output high-quality content, exhibiting excellent instruction following ability. For the few-shot setting, our model exhibits instruction following ability both before and after SFT.

## A.5 Comparison with BitNet

Recently, BitNet [38] introduces a 1-bit model architecture and applies the architecture to train models from scratch, demonstrating the feasibility and application value of the 1-bit model structure. In this paper, we attempt to combine 1-bit quantization with knowledge distillation to quantize the LLaMA-7B model. Unfortunately, despite following the suggestion to use larger learning rates, the behavior remains unstable during training.

Figure 6 shows that the training process of BitNet may suffer from instability during knowledge distillation. We conjecture that it is because the gradient is pretty large when the weight elements fluctuate between +1 and -1, further aggravating the output of the `Linear` layer.

As a more effective measure, the value vectors we propose for quantization not only supplement the necessary floating-point numerical precision but also limit the fluctuation range of the matrix multiplication results after quantization. This can be understood from forward and backward computation, respectively.

**Forward stability.** Quantized matrix multiplication is more prone to overflow than FP16 counterparts in response to minor perturbations of input activations. This is because the magnitude of

Table 5: Instruction following cases in zero-shot and few-shot settings before and after Supervised Fine-Tuning (SFT). The few-shot examples are from MMLU benchmark.

| | |
|---|---|
| **Zero-shot w/o SFT** | **Prompt:**
List several Windows operating systems developed by Microsoft corporation.
**Generation:**
Microsoft Windows 10 is a free operating system developed by Microsoft for use on personal computers and tablets. It was first released to the public on 29 April 2014.
Microsoft Windows 10 is a free operating system developed by Microsoft for use on personal computers and tablets. It was first released to the public on 29 April 2014. Windows 10 is the successor to Windows 8.1.
Microsoft Windows 10 is a free operating system developed by Microsoft for use on personal computers and tablets. It was first released to the public on 29 April 2014. Windows 10 is the successor to Windows 8.1. Windows 10 is the successor to Windows 8.1.
Microsoft Windows 10 is a free operating system developed by Microsoft for use on personal computers and tablets. |
| **Zero-shot w/ SFT** | **Prompt:**
Below is an instruction that describes a task. Write a response that appropriately completes the request.
### Instruction:
List several Windows operating systems developed by Microsoft corporation.
### Response:
**Generation:**
Microsoft has developed several Windows operating systems, including Windows 10, Windows 8, Windows 7, Windows XP, Windows 2000, Windows NT, Windows ME, Windows 95, and Windows 1. Windows 10 is the latest version of the Windows operating system. It was released in 2019. Windows 8 was released in 2011. Windows 7 was released in 2009. Windows XP was released in 2001. Windows NT was released in 1997. Windows ME was released in 1999. Windows 95 was released in 1995. Windows 1 was released in... |
| **Few-shot w/o SFT** | **Prompt:**
The following are multiple choice questions (with answers) about college biology.
Which of the following contain DNA sequences required for the segregation of chromosomes in mitosis and meiosis? A. Telomeres B. Centromeres C. Nucleosomes D. Spliceosomes Answer: B. The Barr body found within female mammalian cells' nuclei is an example of: A. euchromatin B. heterochromatin C. a cistron D. pseudogenes Answer: B. Cleavage symmetry arising from early cell divisions that occur at an oblique angle to the polar axis is known as: A. meroblastic cleavage B. radial cleavage C. spiral cleavage D. superficial cleavage Answer: C. Which of the following is NOT a source of atmospheric carbon? A. Respiration B. Photosynthesis C. Bacterial decomposition D. Combustion of fossil fuels Answer: B. The concept of punctuated equilibrium refers to: A. oscillating ecological successional stages B. ecological succession arrested by sudden environmental changes, e.g., fire C. persistent predator-prey relationships in relatively stable environments D. bursts of speciation followed by relatively unchanging lineages Answer: D. Which of the following is a second messenger that stimulates release of calcium ions into the cytoplasm? A. Prostaglandins B. Calmodulin C. Cyclic AMP D. Inositol triphosphate Answer:
**Generation:**
B |
| **Few-shot w/ SFT** | **Prompt:**
Below is an instruction that describes a task. Write a response that appropriately completes the request.
### Instruction:
The following are multiple choice questions (with answers) about college biology.
Which of the following contain DNA sequences required for the segregation of chromosomes in mitosis and meiosis? A. Telomeres B. Centromeres C. Nucleosomes D. Spliceosomes Answer: B. The Barr body found within female mammalian cells' nuclei is an example of: A. euchromatin B. heterochromatin C. a cistron D. pseudogenes Answer: B. Cleavage symmetry arising from early cell divisions that occur at an oblique angle to the polar axis is known as: A. meroblastic cleavage B. radial cleavage C. spiral cleavage D. superficial cleavage Answer: C. Which of the following is NOT a source of atmospheric carbon? A. Respiration B. Photosynthesis C. Bacterial decomposition D. Combustion of fossil fuels Answer: B. The concept of punctuated equilibrium refers to: A. oscillating ecological successional stages B. ecological succession arrested by sudden environmental changes, e.g., fire C. persistent predator-prey relationships in relatively stable environments D. bursts of speciation followed by relatively unchanging lineages Answer: D. Which of the following is a second messenger that stimulates release of calcium ions into the cytoplasm? A. Prostaglandins B. Calmodulin C. Cyclic AMP D. Inositol triphosphate Answer:
### Response:
**Generation:**
D |

elements in quantized matrices, particularly the value ±1, is far greater than the parameters of most FP16 matrices. By multiplying by value vectors of a magnitude similar to that of the FP16 model, the range of variation in model output activations can be restored to the level of FP16. Furthermore, we also avoid the increasingly large "drift phenomenon" of activations through Post-LayerNorm.

**Backward stability.** Since $\mathrm{Sign}(\cdot)$ function is not differentiable, when the elements of the matrix change, their gradient may become infinite. Similar to forward stability, by multiplying two numerically smaller value vectors, we avoid layer-by-layer accumulation and explosion during gradient back-propagation. Moreover, we implement the derivative function of $\mathrm{Sign}(\cdot)$ using the derivative of the hyperbolic tangent function, thereby avoiding the problem of gradient explosion at the zero point of every weight.

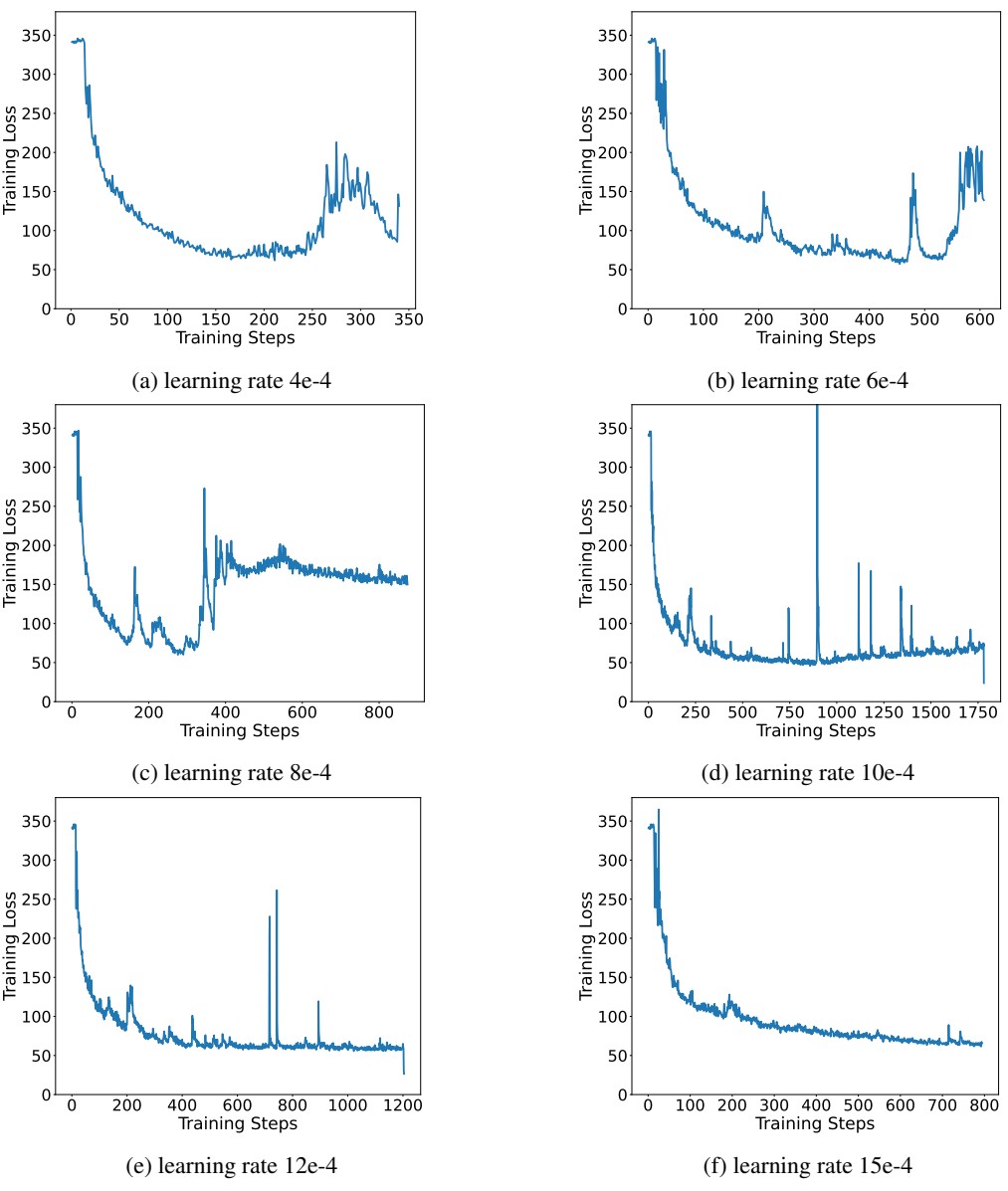

Figure 6: Training comparisons among different learning rates when BitNet performs knowledge distillation from LLaMA-7B. Here we choose the same W1A16 setting as ours. The weight matrices in BitNet are directly copied from the original LLaMA-7B model.

Table 6: Ablation study of different loss on LLaMA-7B. "ATTN" means attention score alignment.

| Loss Setting | Perplexity($\downarrow$) | | Zero-shot Accuracy($\uparrow$) | | | | | | |
| --- | --- | --- | --- | --- | --- | --- | --- | --- | --- |
| | Wiki2 | C4 | Wino. | Hella. | PIQA | BoolQ | ARC-e | ARC-c | Avg. |
| $\mathcal{L}_{\mathrm{KD}}$ | 13.48 | 14.57 | 50.83 | 35.14 | 62.89 | 60.46 | 37.33 | 26.37 | 45.50 |
| $\mathcal{L}_{\mathrm{KD}} + \mathcal{L}_{\mathrm{MSE}}$ ($\alpha = 1$) | **10.19** | **11.40** | 58.48 | **51.54** | **68.01** | 57.28 | **42.47** | **30.20** | 51.33 |
| $\mathcal{L}_{\mathrm{KD}} + \mathcal{L}_{\mathrm{MSE}}$ ($\alpha = 10$) | 10.38 | 11.56 | **60.30** | 50.73 | 67.46 | **62.51** | 41.71 | 29.61 | **52.05** |
| $\mathcal{L}_{\mathrm{KD}} + \mathcal{L}_{\mathrm{MSE}} + \mathcal{L}_{\mathrm{ATTN}}$ | NAN | NAN | - | - | - | - | - | - | - |

### A.6 Discussion on Knowledge Distillation

Although knowledge distillation is not the main contribution of this paper, we nevertheless provide the rationale behind certain settings used in our experiments to explain the necessity of these configurations.

We firstly explain the role of different loss functions in guiding the process of knowledge transfer. Fundamentally, distillation loss alone can achieve a satisfactory transfer process (comparing to other baselines). Additionally, as shown in the Table 6, aligning the hidden states between layers can result in a quantized model with better perplexity. However, further incorporating attention score alignment on this basis leads to the model failing to converge. LLM-QAT [21] has conducted similar experiments on quantization-aware knowledge distillation loss and concluded that using only the distillation loss yields the best results. The difference in conclusions may stem from two factors. On one hand, due to our adoption of a novel model architecture, which differs from theirs, the optimal usage of loss functions may be different as well. On the other hand, as we focus on extremely low bit-width compression, each layer of the model suffers significant information loss compared to the teacher model. The regularization of hidden states between layers may help reduce the variance in the learning process, thus demonstrating stronger generalization.

Furthermore, we also discuss the cost of our quantization method. Using LLaMA-7B as an example, quantizing the model with our method requires approximately 7 days on 8 A100-80GB GPUs. In comparison, training the LLaMA-7B model from scratch consumes 82,432 GPU hours [34]. The quantization time, being less than 2% of the pretraining time, is still an acceptable cost.

### A.7 Average Bit-width of Linear Layer

This subsection formulates the calculation of the average bit-width of `Linear` layers. Assume there is a weight matrix with a shape of $4096 \times 4096$ in such a layer, the number of bits in every component is

$$1 \times 4096 \times 4096,$$
$$16 \times 1 \times 4096 \times 2,$$

where the first is for the 1-bit quantized weight matrix and the second is for the two FP16 value vectors. Hence the overall number of bits is $16,908,288$. Moreover, the number of parameters is $4096 \times 4096 + 2 \times 4096 \times 1 = 16,785,408$. Therefore, the average bit-width of this `Linear` layer is $16,908,288 \div 16,785,408 \approx 1.0073$.

## B Limitations

Although our proposed method significantly reduces the memory footprint of LLMs, bringing hope for efficient deployment of them, there are still some limitations. Firstly, compared to the original model, our extremely low-bit quantization inevitably incurs a performance loss. Additionally, we are yet to understand the mathematical principles behind the optimal parameters of the 1-bit quantized model, thus capability transfer can only be achieved through the relatively costly process of KD. Fortunately, this cost is a one-time expense. Moreover, due to the unique nature of 1-bit quantization, our method can not be naturally extended to higher bit-width. Lastly, we have not considered the activation quantization and leave it as future work.

# C   Ethics Statement

In this study, we employ models that are publicly available and open source. We affirm that the use of these models aligns with their original intended purposes. These models have been utilized strictly within the scope of academic and research-based activities, adhering to ethical guidelines and ensuring compliance with open-source licenses.

