# OpenReview forum: "OneBit: Towards Extremely Low-bit Large Language Models"
_NeurIPS.cc/2024/Conference — NeurIPS 2024 poster_

### Official Review · Reviewer_a9Ag · 2024-06-25

**Soundness:** 3
**Presentation:** 3
**Contribution:** 3
**Rating:** 7
**Confidence:** 4

**Summary:**

This paper presents OneBit, a framework for quantizing large language models (LLMs) to 1-bit weight matrices. Unlike existing methods that rely on 4-bit or 8-bit quantization to avoid severe performance degradation, OneBit introduces a novel 1-bit parameter representation and an effective parameter initialization method based on matrix decomposition.

**Strengths:**

- This paper proposes an aggressive compression method, exploring the feasibility and challenges of compressing LLMs to 1-bit. This is a highly meaningful research direction, and similar work should be encouraged for publication. However, I have some concerns about this approach, which I will detail in the questions section.

- The paper is well-organized and well-written.

- The experiments compare the proposed method with current popular low-bit quantization methods and demonstrate superior results. Additionally, the experiments address an important question in the current LLM research field: whether to use quantized large models or directly use smaller models.

**Weaknesses:**

- Entropy Encoding: Intuitively, 1-bit compressed models should be suitable for entropy encoding. Previous work has demonstrated that quantized LLMs still have compressibility [1,2]. Have the authors tried using popular entropy encoders to further compress these weights?

- Inference Calculations: During inference, do g and h participate in the calculations? Does this mean that 1-bit net is not entirely integer-based computation? Does each layer have its own G and h?

- Dequantization: Is dequantization required between layers during inference?

- Knowledge Distillation (KD): In line 191, it is mentioned that KD does not use LM loss. Why is that?

- Comparison with 1.58bitnet: Have the authors considered comparing their method with 1.58bitnet [3]?

- Table 2 Details: Does Table 2 show the performance of W1A16 or W2A16? If it is W2A16, where is the performance of W1A16 reported? Why are there no experimental results for 16B and 60B models in Table 2? I suspect that extreme quantization has a more detrimental effect on larger models.

- Comparison in Figure 3: Figure 3 compares the performance of 1-bit quantized 7B models with smaller models. Are these smaller models in full precision? How would the results compare with 8-bit smaller models?

- Code Availability: Will the code be open-sourced?

- Minor Issues: Figures 3 and 4 are not clear when printed in gray-scale.

References:

[1] Mao Y, Wang W, Du H, et al. On the compressibility of quantized large language models. arXiv preprint arXiv:2403.01384, 2024.

[2] [2] Hershcovitch M, Choshen L, Wood A, et al. Lossless and Near-Lossless Compression for Foundation Models[J]. arXiv preprint arXiv:2404.15198, 2024.
[3] Ma S, Wang H, Ma L, et al. The Era of 1-bit LLMs: All Large Language Models are in 1.58 Bits[J]. arXiv preprint arXiv:2402.17764, 2024.

**Questions:**

See above.

**Limitations:**

See above.

---

> ### Author Rebuttal · Authors · 2024-08-04
>
> We appreciate the time and effort you have invested in reviewing our paper.
>
> **Question 1**: "Have the authors tried using popular entropy encoders to further compress these weights?"
>
> This maybe a great intuition! Theoretically, a model quantized to 1-bit does have the potential for further compression using entropy coding. In fact, we are currently exploring this possibility, and it holds independent research value.
>
> **Question 2**: "During inference, do g and h participate in the calculations? Does this mean that 1-bit net is not entirely integer-based computation? Does each layer have its own G and h?"
>
> Yes. The FP16 vectors g/h participate in the calculation during inference. Our proposed 1-bit net structure is indeed not entirely integer-based. The core of our idea is the minimal use of floating-point and their substantial benefits. Finally, each Linear layer has its own G and h.
>
> **Question 3**: "Is dequantization required between layers during inference?"
>
> No. Dequantization is not required between layers during inference, because the original weight have been entirely converted into a quantized matrix W and two floating-point vectors g/h.
>
> **Question 4**: "Why is it mentioned that KD does not use LM loss in L191?"
>
> We specifically mention in the paper that LM loss is not used to ensure accurate description and facilitate reproducibility of our results. The reason for not using LM loss is that this loss does not demonstrate beneficial effects in our experiments.
>
> **Question 5**: "Have the authors considered comparing their method with Bitnet-b1.58?"
>
> BitNet-b1.58 is a work slightly later than ours, aiming to propose a possible 1-2 bit model structure (examining the performance of training from scratch), which differs from model quantization that transforms a "pre-trained model" into a low-bit representation. However, we are also curious whether the BitNet-(1b/1.58b) structure can be used for quantizing existing models, and **we have discussed the relevant results of BitNet-1b in Appendix A.5 and Fig. 6**. BitNet-b1.58 is also similar to this. The conclusion shows that, compared to BitNet, our model structure achieves stable ability transfer and training process, whereas BitNet-(1b/1.58b) fails to converge during quantization-aware distillation.
>
> **Question 6**: "**1.** Does Table 2 show the performance of W1A16 or W2A16? If it is W2A16, where is the performance of W1A16 reported? **2.** Why are there no experimental results for 16B and 60B models in Table 2? I suspect that extreme quantization has a more detrimental effect on larger models."
>
> 1. Table 2 compares our method with four baselines: FP16, GPTQ, LLM-QAT, and OmniQuant. FP16 represents the non-quantized model, serving as the upper bound for all methods’ capabilities. As the first 1-bit weight quantization method, our approach, OneBit, uses W1A16 quantization. The baselines GPTQ, LLM-QAT, and OmniQuant use a W2A16 quantization level.
>
> 2. Due to the **limitation of computational resources**, we are currently unable to perform experiments on larger scale models, although we have been trying to get the resources. However, our existing results **indicate an exciting trend**: "the larger the model, the smaller the performance gap between the FP16 precision and the W1A16 quantized model." This conclusion is mentioned in lines **L238~L240** of our paper.
>
> **Question 7**: "Figure 3 compares the performance of 1-bit quantized 7B models with smaller models. Are these smaller models in full precision? How would the results compare with 8-bit smaller models?"
>
> Yes, the smaller models are in full FP16 precision. The conclusion is that our 1-bit quantized model is better than the similar scaled full precision model. As the upperbound, the full precision model is better than the same 8-bit smaller models. Hence, we can directly conclude that our 1-bit quantized model is better than the 8-bit smaller models.
>
> **Question 8**: "Code Availability: Will the code be open-sourced?"
>
> We have submitted the core OneBit Linear Layer python code in the supplemental material to NeurIPS. Additional, all the code, data, and checkpoints are fully open-sourced on other platforms. They will be made completely public after the peer review process is concluded.
>
> **Question 9**: "Minor Issues: Figures 3 and 4 are not clear when printed in gray-scale."
>
> Thank you for your reminder and suggestions! We will reconsider their line styles, colors, and plotting methods, and make adjustments in the revised version.
>
> We look forward to hearing from you and hope to address any concerns you may have about our work. Please let us know if you have any further questions.

---

> > ### Comment · Reviewer_a9Ag · 2024-08-12
> >
> > Thank you for the clarification and new sensitivity analysis. I will keep my score.

---

> > > ### Author Response · Authors · 2024-08-12
> > >
> > > Thank you very much for your continued review. We are grateful for your positive feedback and the time you have devoted to evaluating our work. Please don’t hesitate to reach out if there are any further questions or points of discussion. We remain at your disposal for any additional clarifications.

---

### Official Review · Reviewer_mFhw · 2024-07-08

**Soundness:** 3
**Presentation:** 3
**Contribution:** 3
**Rating:** 5
**Confidence:** 4

**Summary:**

This paper explores an innovative 1-bit quantization framework for Large Language Models (LLMs) to significantly reduce their memory and computational demands. Traditional methods face severe performance drops with reduced bit-width; however, this paper introduces a novel quantization and initialization approach that maintains at least 81% of the original model’s performance, even with extreme bit-width reduction. The proposed method, which includes a specialized matrix decomposition and parameter initialization, demonstrates strong performance and robustness in experiments, establishing a new direction for deploying LLMs on resource-constrained environments.

**Strengths:**

The paper is easy to follow. The proposed method shows good performance, achieving at least 81% of the unquantized model’s efficacy, a significant achievement given the drastic reduction in model complexity and size.

**Weaknesses:**

A significant issue discussed in the paper is the lack of a specialized CUDA kernel for optimizing binary operations, which hinders accurate evaluation of the additional computational costs associated with the two FP vectors $\mathbf{a}$ and $\mathbf{b}$. This limitation complicates the assessment of their impact on overall performance. Furthermore, despite the inclusion of $\mathbf{a}$ and $\mathbf{b}$, there remains a considerable performance decline compared to FP16 models, challenging the practical applicability of this approach in real-world settings.

**Questions:**

1. The experiments described in Section 4.3 may not provide an appropriate comparison. Assessing the proposed method alongside directly training smaller models or utilizing low-rank decomposition to minimize parameter counts involves fundamentally different approaches to reducing model size. Additionally, the proposed method incorporates knowledge distillation, which is not employed in the baseline methods being compared, potentially skewing performance comparisons.

2. An essential ablation study is notably absent from the discussion. The proposed method incorporates two additional floating-point vectors, $\mathbf{a}$ and $\mathbf{b}$, for binary quantization. Yet, the impact of these vectors on performance enhancement remains unclear, highlighting a gap in the evaluation of the method’s effectiveness.

3. To offer a more comprehensive evaluation, I recommend including assessments on generative tasks, such as code generation. This would provide deeper insights into the versatility and practical applicability of the proposed method across different domains.

**Limitations:**

The authors have discussed the limitations of their study.

---

> ### Author Rebuttal · Authors · 2024-08-04
>
> We appreciate the time and effort you have invested in reviewing our paper.
>
> **Weakness**: "It might lack **specialized CUDA kernel** for optimizing binary operations and the additional computational costs associated with the two FP vectors a and b may be not clear. Moreover, the **performance decline** challenging the practical applicability in real-world settings."
>
> - For CUDA kernel and inference time:
>
> The potential advantage of our method benefits from a special multiplication, **INT1(W) * FP16(A)**, in which the traditional FP16*FP16 can be quickly and efficiently replaced by setting the sign bit of FP16 activation. For example, we have **W=(0.1, -0.3, -0.2)** and **A=(0.8, 0.2, 0.7)**, and traditional W*A maybe 0.1*0.8+(-0.3)*0.2+(-0.2)*0.7 and it can be quantized to Pos(0.8)+Neg(0.2)+Neg(0.7). Here **Pos(·)** represents the machine instruction for setting the sign bit of a floating-point number to '+', while **Neg(·)** represents the opposite. Unfortunately, since this computation method is **not yet perfectly supported at the GPU hardware level**, we cannot demonstrate how fast OneBit truly is on the device by carefully designing CUDA kernel.
>
> All experiments have been conducted using FP16 format containing only ±1 for simulation. We provide an possible efficient implementation of the OneBit Linear Layer in the supplemental material (with parallel tensor computation). If we **simply simulate INT1 using FP16 format**, the inference time is approximately **1.2x** that of the original model. In fact, thanks to the **broadcasting mechanism** of tensors, the element-wise multiplication of matrices and vectors can be performed very quickly. Therefore, the **additional vector incurs minimal time overhead**. If we **aim for extreme space compression**, i.e., expanding the compressed weight during inference, the inference latency is approximately **2.2x** that of the original model. It is worth noting that **most weight-only quantization studies introduce additional inference latency**. Additionally, we sincerely hope that our work, along with recent similar efforts, will encourage device providers to support this faster computation method at the hardware level.
>
> - For practical applicability:
>
> Even though our proposed quantization is lossy compression, we strive to demonstrate its practical value in this paper. In Section 4, we show our method’s **excellent performance on benchmarks** by comparing it with strong baselines. In Appendix A.4, we demonstrate the practicality of our method through SFT and **instruction-following tests**. Moreover, compared to contemporaneously published method [1], our approach proves to be superior in both effectiveness and capability, underscoring the research and practical value of our method.
>
> **Question 1**: "The proposed quantization method and training smaller models or utilizing low-rank decomposition are different approaches, and their comparison may not appropriate. Additionally, comparing with the baselines, which not employe knowledge distillation, may be appropriate as well."
>
> We understand the reviewer's concern. It is important to clarify that our comparisons with these different compression methods are **not intended to defeat them**, but rather to **demonstrate the model’s capabilities from another perspective**—specifically, that it **performs better than smaller models of the similar scale**. Therefore, we did not include this comparison as the main result in Sec 4.2, but rather **as a separate subsection**, "Problem Solving Ability".
>
> Additionally, regarding the second concern, firstly it is crucial for us to **compare with strong baselines in model quantization, regardless of the methods they employ**. Among the baselines we selected, **there are also methods based on knowledge distillation (training)**, such as LLM-QAT [2], which was once a strong baseline. **Please refer to Sections 4.1 & 4.2.**
>
> **Question 2**: "An essential ablation study is notably absent from the discussion. The impact of these vectors a/b on performance enhancement remains unclear."
>
> In fact, the **comparison with BitNet is essentially an ablation study** concerning the a/b vectors, as the **main difference** between our model structure and BitNet lies in the introduction of the FP16 a/b vectors. Due to space limitation, the main discussion is placed in Appendix A.5, with the conclusions only presented in Sections 5.2 & 5.3 of the main text. We will address this issue in the revised version. From the discussion in A.5, we demonstrate that, **once these two vectors are deleted (BitNet), the 1-bit weight W-only model fails to converge during quantization-aware distillation.** Hence, our model structure (with a/b vectors) achieves stable ability transfer and training process, which demonstrate its necessity.
>
> **Question 3**: "Including generative tasks would provide deeper insights into the versatility and practical applicability of the proposed method across different domains."
>
> Thank you for your valuable suggestions! We will consider adding more examples of generative tasks in the revised version's Appendix to demonstrate the practical value of our method.
>
> We look forward to hearing from you and hope to address any concerns you may have about our work. Please let us know if you have any further questions.
>
> Reference:
>
> [1] Huang W, Liu Y, Qin H, et al. BiLLM: Pushing the Limit of Post-Training Quantization for LLMs[C]//Forty-first International Conference on Machine Learning. 2024.
>
> [2] Liu Z, Oguz B, Zhao C, et al. LLM-QAT: Data-free quantization aware training for large language models[J]. arXiv preprint arXiv:2305.17888, 2023.

---

> ### Comment · Reviewer_mFhw · 2024-08-12
> **Official comments by Reviewer mFhw**
>
> Thank you to the authors for rebuttal and the clarifications provided. Based on your responses, I remain inclined to keep the score.

---

> > ### Author Response · Authors · 2024-08-12
> >
> > Thank you very much for your continued review. We are grateful for your positive feedback and the time you have devoted to evaluating our work. Please don’t hesitate to reach out if there are any further questions or points of discussion. We remain at your disposal for any additional clarifications.

---

### Official Review · Reviewer_gg38 · 2024-07-10

**Soundness:** 2
**Presentation:** 3
**Contribution:** 2
**Rating:** 5
**Confidence:** 4

**Summary:**

This paper proposes OneBits, a novel quantization-aware training methodology for 1-bit large language models (LLMs). OneBits introduces two key contributions for training 1-bit models. First, it presents a new 1-bit binary quantization linear design that separates the weight matrix into sign and value components. The sign is packed into INT1, while the value is decomposed using a 1-rank decomposition factor added to the linear operation. Second, to train the 1-bit models in the linear layers of BitNets, OneBits modifies the traditional quantization-aware training (QAT) method by augmenting the cross-entropy loss function with an additional term for the reconstruction error of each layer, resulting in the final objective loss function.

Using the proposed approach, OneBits is applied to various decoder-only LLM models. The comparisons between OneBits (W1A16) and other methods like LLM-QAT, AWQ, and OmniQuant (W2A16) demonstrate that OneBits achieves superior performance in common sense reasoning tasks.

**Strengths:**

- The paper proposes a final objective loss function that combines the final cross-entropy loss with the reconstruction error of each layer using a Quantization-aware Knowledge Transfer method. The effectiveness of incorporating the reconstruction error is demonstrated through an ablation study (Table 6).
- Unlike the traditional 1-bit linear design in BitNet, the authors introduce a new 1-bit binary quantization linear design that includes scaling factors (g/h) for each input/output channel of the weight matrix. They also propose an initialization method from a pretrained model using Sign-Value Independent Decomposition (SVID).
- To initialize the scaling factors (g/h) for each input/output channel, the paper explores various 1-rank decomposition methods for value in SVID, including Singular Value Decomposition (SVD) and Non-negative Matrix Factorization (NMF). Experimental results indicate that the 1-rank decomposition of value using NMF is more effective than SVD.

**Weaknesses:**

- While the paper demonstrates zero-shot performance in terms of PPL and CSR, it lacks experiments on how the same model maintains performance in few-shot scenarios, such as the MMLU benchmark.
- If the quality of this generated data is poor, it could negatively impact the performance of the OneBit LLM. The paper does not clearly explain why self-generated data was used instead of public datasets like C4.
- The analysis of OneBit LLM's benefits in terms of inference latency and throughput relative to accuracy is insufficient. A detailed examination of these metrics would provide a more comprehensive understanding of the advantages of using OneBit LLM.

**Questions:**

- While the OneBit method has demonstrated independent evaluation of zero-shot and few-shot performance, showing effectiveness compared to LLM-QAT and OmniQuant, it does not provide evidence of a single OneBit model performing well in both zero-shot and few-shot scenarios simultaneously. It would be valuable to present combined performance results, such as including MMLU results in Table 2 for a comprehensive comparison.
- When comparing the quality of output generated by OneBit LLM to other models using metrics like AlpacaEval, what trends or patterns emerge regarding the quality of generated data?

**Limitations:**

Since OneBit LLM is applied only to weights, this method is likely to be effective in improving latency and throughput, particularly in scenarios involving small batch sizes during the generation phase. In this paper, activation quantization has not been considered, and further research in this area is necessary to optimize performance.

---

> ### Author Rebuttal · Authors · 2024-08-04
>
> We appreciate the time and effort you have invested in reviewing our paper.
>
> **Weakness 1**: "It might lack a few-shot benchmark result such as MMLU."
>
> Although few-shot evaluation is not a necessary component in most model quantization research [1,2,3], we still evaluated the **5-shot** performance of OneBit-7B on MMLU in Sec. 4.3. **Please refer to Sec. 4.3 and Fig. 3(b).**
>
> **Weakness 2**: "Poor self-generated data may negatively impact the performance of OneBit. The reason for using it instead of public dataset C4 is not clearly explained."
>
> Here we **follow LLM-QAT (L204)** to perform knowledge distillation. In LLM-QAT [5], the authors **have demonstrated the effectiveness of using self-generated data**, which **provide a comprehensive coverage of sample-able tokens**. Using external data may introduce **bias**. In fact, in terms of content, the quality of self-generated data (which we have open-sourced on other platforms) is indeed inferior to the carefully cleaned real data. However, it can **maximize the ability transfer of the FP16 teacher in the student model**. We take Llama-7B and PPL as examples to compare the effects of the two types of data, where C4 has the same amount of data after sampling as the self-generated data.
>
> | data source | Wiki2 | C4 |
> | -- | -- | -- |
> | sampled-C4 | 15.01 | 12.29 |
> | self-generated | **10.19** | **11.40** |
>
> **Weakness 3**: "A detailed examination of inference latency may be insufficient."
>
> The potential advantage of our method benefits from a special multiplication, **INT1(W) * FP16(A)**, in which the traditional FP16*FP16 can be quickly and efficiently replaced by setting the sign bit of FP16 activation. For example, we have **W=(0.1, -0.3, -0.2)** and **A=(0.8, 0.2, 0.7)**, and traditional W*A maybe 0.1*0.8+(-0.3)*0.2+(-0.2)*0.7 and it can be quantized to Pos(0.8)+Neg(0.2)+Neg(0.7). Here **Pos(·)** represents the **machine instruction** for setting the sign bit of a floating-point number to '+', while **Neg(·)** represents the opposite. Unfortunately, since this computation method is **not yet perfectly supported at the GPU hardware level**, we cannot demonstrate how fast OneBit truly is on the device. All experiments have been conducted using FP16 format containing only ±1 for simulation. We provide an possible efficient implementation of the OneBit Linear Layer in the supplemental material. If we **simply simulate INT1 using FP16 format**, the inference time is approximately **1.2x** that of the original model. If we **aim for extreme space compression**, i.e., expanding the compressed weight during inference, the inference latency is approximately **2.2x** that of the original model. It is worth noting that **most weight-only quantization studies introduce additional inference latency**. Additionally, we sincerely hope that our work, along with recent similar efforts, will encourage device providers to support this faster computation method at the hardware level.
>
> **Question 1**: "It would be valuable to present combined performance results, such as including MMLU results in Table 2."
>
> Thank you for your suggestion! We did not take this for 2 reasons: first, other than OmniQuant, the W2A16 baselines perform poorly on MMLU. Second, we wanted to compare our method’s performance in general knowledge using MMLU, hence we included it in Section 4.3.
>
> **Question 2**: "What trends or patterns emerge regarding the quality of generated data comparing to other models?"
>
> As shown in PPL (Tab.2) and the content in Tab. 5, OneBit can fluently output content as long as it has not forgotten the knowledge in that domain, showing no significant difference from the original model. However, once OneBit forgets the knowledge of a certain domain, it tends to output a minimal number of tokens followed by '\n', and then stops outputting.
>
> **Limitation**: "Activation quantization has not been considered, and further research in this area is necessary to optimize performance."
>
> To date, weight quantization [1,4] and weight-activation[2,3,5] quantization **remain 2 distinct research paths**. The reason and difficulty lie in the fact that activation quantization also compromises the model’s capabilities, with significant compression of activations causing severe degradation of the model’s performance. Therefore, the strongest baseline for W-quantization is currently W1A16, while WA-quantization can generally achieve W4A4. We are working towards further compressing activations, but we think that not considering activations may not be a limitation.
>
> We look forward to hearing from you and hope to address any concerns you may have about our work. Please let us know if you have any further questions.
>
> Reference:
>
> [1] Frantar E, Ashkboos S, Hoefler T, et al. OPTQ: Accurate quantization for generative pre-trained transformers[C]//The Eleventh International Conference on Learning Representations. 2022.
>
> [2] Xiao G, Lin J, Seznec M, et al. Smoothquant: Accurate and efficient post-training quantization for large language models[C]//International Conference on Machine Learning. PMLR, 2023: 38087-38099.
>
> [3] Shao W, Chen M, Zhang Z, et al. OmniQuant: Omnidirectionally Calibrated Quantization for Large Language Models[C]//The Twelfth International Conference on Learning Representations. 2024.
>
> [4] Lin J, Tang J, Tang H, et al. AWQ: Activation-aware Weight Quantization for On-Device LLM Compression and Acceleration[J]. Proceedings of Machine Learning and Systems, 2024, 6: 87-100.
>
> [5] Liu Z, Oguz B, Zhao C, et al. LLM-QAT: Data-free quantization aware training for large language models[J]. arXiv preprint arXiv:2305.17888, 2023.

---

> > ### Author Response · Authors · 2024-08-12
> > **Invitation to Participate in the Discussion Period**
> >
> > Thank you very much for your review. We have provided detailed responses to your question. If you could participate in the discussion period, we would be very grateful again.

---

> > ### Comment · Reviewer_gg38 · 2024-08-13
> > **Response from Reviewer gg38**
> >
> > Thank you for your considerate response. While most of my concerns have been addressed, I still believe it is important to examine how the gap between Zero-shot and Few-shot performance changes before and after applying OneBits to the public LLM models in Table 2.
> > When performing QAT from scratch, as with OneBits-7B, I believe that training with at least a similar number of tokens to what is suggested by the Chinchilla-optimal is necessary to observe a reliable trend.
> > However, considering that Figures 3(a) and 3(b) show that OneBit-7B achieves performance comparable to the 1B-scale model, which was trained with a larger amount of data, and that OneBit demonstrates a meaningful performance improvement over other quantization methods on public LLMs, I have decided to raise my score from 4 to 5.

---

> > > ### Author Response · Authors · 2024-08-13
> > >
> > > Thank you very much for your continued review. We are grateful for your positive feedback and the time you have devoted to evaluating our work. Please don’t hesitate to reach out if there are any further questions or points of discussion. We remain at your disposal for any additional clarifications.

---

### Official Review · Reviewer_9tLG · 2024-07-14

**Soundness:** 3
**Presentation:** 3
**Contribution:** 3
**Rating:** 6
**Confidence:** 4

**Summary:**

This paper proposes OneBit, which quantizes the LLM weight matrices to 1-bit and achieves good performance and improved convergence speed by using two additional vectors with FP16 per one linear layer.

**Strengths:**

1. This paper is generally well-written and easy to follow.

2. The memory required for the model part is less than other methods.

3. Their methods nicely outperforms other methods in many tasks.

**Weaknesses:**

1. It will be great if they compare with other one-bit based quantization methods such as BitNet.

2. I recommend authors to add empirical results for larger models like Llama-70b.

**Questions:**

1. Would it be possible to compare the inference speed of this method and other methods? I am curious about the potential delay in inference caused by using additional FP16 vectors.

---

> ### Author Rebuttal · Authors · 2024-08-04
>
> We appreciate the time and effort you have invested in reviewing our paper.
>
> **Weakness 1**: "It will be great if they compare with other one-bit based quantization methods such as BitNet."
>
> Converting the **"pre-trained model"** into a low-bit representation is **the focus of almost all research on model quantization** [1,2,3], and we also follow this point. BitNet, as another well-known work, differs from model quantization by proposing a possible model structure with 1-bit weights (they focus on training low-bit model from scratch). Therefore, BitNet and our model quantization essentially address different research problems. In fact, we are also curious whether the BitNet structure can be used for quantizing existing models, and **we have discussed the relevant results in Appendix A.5 and Fig. 6**. The conclusion shows that, compared to BitNet, our model structure achieves stable ability transfer and training process, whereas BitNet fails to converge during quantization-aware distillation. Hence, we cannot provide benchmark results of BitNet in the main text.
>
> **Weakness 2**: "I recommend authors to add empirical results for larger models like Llama-70b."
>
> Due to the **limitation of computational resources**, we are currently unable to perform experiments on 70B-scale models, although we have been trying to get the resources. However, our existing results **indicate an exciting trend**: "the larger the model, the smaller the performance gap between the FP16 precision and the W1A16 quantized model." This conclusion is mentioned in lines **L238~L240** of our paper.
>
> **Question**: "Would it be possible to compare the inference speed of this method and other methods? I am curious about the potential delay in inference caused by using additional FP16 vectors."
>
> It is possible to compare the inference speed of ours **with FP16 baseline**. In a 4k-length inference test, if the weights are simulated in FP16 format containing only ±1, the OneBit model, which includes FP16 vectors, takes approximately 1.2x the duration of Llama-7b without these vectors. In fact, thanks to the **broadcasting mechanism** of tensors, the element-wise multiplication of matrices and vectors can be performed very quickly. Therefore, **the additional vector incurs minimal time overhead**.
>
> We look forward to hearing from you and hope to address any concerns you may have about our work. Please let us know if you have any further questions.
>
> Reference:
>
> [1] Frantar E, Ashkboos S, Hoefler T, et al. OPTQ: Accurate quantization for generative pre-trained transformers[C]//The Eleventh International Conference on Learning Representations. 2022.
>
> [2] Xiao G, Lin J, Seznec M, et al. Smoothquant: Accurate and efficient post-training quantization for large language models[C]//International Conference on Machine Learning. PMLR, 2023: 38087-38099.
>
> [3] Liu Z, Oguz B, Zhao C, et al. LLM-QAT: Data-free quantization aware training for large language models[J]. arXiv preprint arXiv:2305.17888, 2023.

---

> > ### Comment · Reviewer_9tLG · 2024-08-11
> >
> > Thank you for your rebuttal and clarification. I remain inclined to accept this work and will maintain my score of 6.

---

> > > ### Author Response · Authors · 2024-08-12
> > >
> > > Thank you very much for your continued review. We are grateful for your positive feedback and the time you have devoted to evaluating our work. Please don’t hesitate to reach out if there are any further questions or points of discussion. We remain at your disposal for any additional clarifications.

---

### Decision · Program_Chairs · 2024-09-25

**Decision:**

Accept (poster)

**Comment:**

This paper introduces an aggressive quantization technique that is of interest to the community. However, I would recommend the authors moderate some of their claims. For instance, it is not, strictly speaking, 1-bit quantization when considering the two vector values.